# MLMODELSCOPE: A DISTRIBUTED PLATFORM FOR MODEL EVALUATION AND BENCHMARKING AT SCALE

## ABSTRACT

Machine Learning (ML) and Deep Learning (DL) innovations are being introduced at such a rapid pace that researchers are hard-pressed to analyze and study them. The complicated procedures for evaluating innovations, along with the lack of standard and efficient ways of specifying and provisioning ML/DL evaluation, is a major "pain point" for the community. This paper proposes MLModelScope, an open-source, framework/hardware agnostic, extensible and customizable design that enables repeatable, fair, and scalable model evaluation and benchmarking. We implement the distributed design with support for all major frameworks and hardware, and equip it with web, command-line, and library interfaces. To demonstrate MLModelScope's capabilities we perform parallel evaluation and show how subtle changes to model evaluation pipeline affects the accuracy and HW/SW stack choices affect performance.

## 1 INTRODUCTION

The emergence of Machine Learning (ML) and Deep Learning (DL) within a wide array of application domains has ushered in a great deal of innovation in the form of new models and hardware/software (HW/SW) stacks (frameworks, libraries, compilers, and hardware accelerators) to support these models. Being able to evaluate and compare these innovations in a timely manner is critical for their adoption. These innovations are introduced at such a rapid pace (Dean et al., 2018; arXiv ML Papers Statistics) that researchers are hard-pressed to study and compare them. As a result, there is an urging need by both research and industry for a *scalable* model/HW/SW evaluation platform.

Evaluation platforms must maintain *repeatability* (the ability to reproduce a claim) and *fairness* (the ability to keep all variables constant and allow one to quantify and isolate the benefits of the target of interest). For ML/DL, repeatable and fair evaluation is challenging, since there is a tight coupling between model execution and the underlying HW/SW components. Model evaluation is a complex process where the model, dataset, evaluation method, and HW/SW stack must work in unison to maintain the accuracy and performance claims (e.g. latency, throughput, memory usage). To maintain repeatability, authors are encouraged to publish their code, containers, and write documentation which details the usage along with HW/SW requirements (Mitchell et al., 2019; Reproducibility Checklist; Dodge et al., 2019; Lipton & Steinhardt, 2019; Pineau et al., 2018). Often, the documentation miss details which make the results not reproducible. To perform a fair evaluation, evaluators have to manually normalize the underlying stack and delineate the codes to characterize performance or accuracy. This is a daunting endeavor. As a consequence, repeatable and fair evaluation is a "pain-point" within the community (Gundersen et al., 2018; Plesser, 2018; Ghanta et al., 2018; Hutson, 2018; Li & Talwalkar, 2019; Tatman et al., 2018; Reproducibility in Machine Learning; ICLR Reproducibility Challenge). Thus, an evaluation platform design must have a standard way to specify, provision, and introspect evaluations to guarantee repeatability and fairness.

In this paper, we propose MLModelScope: a distributed design which consists of a specification and a runtime that enables repeatable, fair, and scalable evaluation and benchmarking. The proposed specification is a text-based and encapsulates the model evaluation by defining its pre-processing, inference, post-processing pipeline steps and required SW stack. The runtime system uses the evaluation specification along with user-defined HW constraints as input to provision the evaluation, perform benchmarking, and generate reports. More specifically, MLModelScope guarantees repeatable and fair evaluation by (1) defining a novel scheme to specify model evaluation which

separates the entanglement of data/code/SW/HW; (2) defining common techniques to provision workflows with specified HW/SW stacks; and (3) providing a consistent benchmarking and reporting methodology. Through careful design, MLModelScope solves the design objectives while being framework/hardware agnostic, extensible, and customizable.

In summary, this paper makes the following contributions: ① we comprehensively discuss the complexity of model evaluation and describe prerequisites for a model evaluation platform. ② We propose a model evaluation specification and an open-source, framework/hardware agnostic, extensible, and customizable distributed runtime design which consumes the specification to execute model evaluation and benchmarking at scale. ③ We implemented the design with support for Caffe, Caffe2, CNTK, MXNet, PyTorch, TensorFlow, TensorRT, and TFLite, running on ARM, Power, and x86 with CPU, GPU, and FPGA. ④ For ease of use, we equip MLModelScope with command line, library, and ready-made web interfaces which allows "push-button" model evaluation*. ⑤ We also add introspection capability in MLModelScope to analyze accuracy at different stages and capture latency and memory information at different levels of the HW/SW stack. ⑥ We showcase MLModelScope by running experiments which compare different model pipelines, hardware, and frameworks.

## 2 Model Evaluation Challenges

Model evaluation is complex. Researchers that publish and share DL models can attest to that but are sometimes unaware of the full scope of this complexity. To perform repeatable and fair evaluation, we need to be cognizant of the HW/SW stack and how it affects the accuracy and performance of a model. Figure 1 shows our classification of the HW/SW stack levels. **Model level** (L1) evaluates a model by performing input pre-processing, model inference, and post-processing. The pre-processing stage

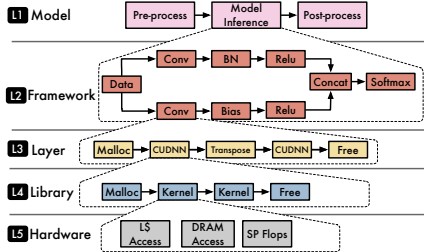

Figure 1: Execution of a model evaluation at different levels of hardware and software abstractions on GPUs.

transforms the user input into a form that the model expects. The model inference stage calls the framework's inference API on the processed input and produces an output. The post-processing stage transforms the model output to a form that can be viewed by a user or used to compute metrics. **Framework level** (L2) performs model inference by executing the layers in the model graph using a framework such as TensorFlow, MXNet, or PyTorch. **Layer level** (L3) executes a sequence of ML library calls for layers such as convolution, normalization, or softmax. **ML Library level** (L4) invokes a chain of system library calls for functions in ML libraries such as cuDNN(Chetlur et al., 2014), MKL-DNN (MKL-DNN) or OpenBLAS (Xianyi et al., 2014). And, last but not the least, at the **hardware level** (L5), there are CPU/GPU instructions, disk, and network I/O events, and other low-level system operations through the entire model evaluation. All the HW/SW abstractions must work in unison to maintain the reported accuracy and performance claims. When things go awry, each level within the abstraction hierarchy can be suspect.

Currently, model authors distribute models by publishing documentation and ad hoc scripts to public repositories such as GitHub. Due to the lack of specification, authors may under-specify or omit key aspects of model evaluation. This inhibits, or makes it difficult, for others to repeat their evaluations or validate their claims. Thus all aspects of the model evaluation must be captured by a evaluation platform to guarantee repeatability. To highlight this, consider the model evaluation pipeline at L1. While the model inference stage is relatively straight forward, the pre- and post-processing stages are surprisingly subtle and can easily introduce discrepancies in the results. Some of the discrepancies might be "silent errors" — where the evaluation is correct for the majority of the inputs but is incorrect for a small number of cases. In general, accuracy errors due to under-specifying pre- and post-processing are difficult to identify and even more difficult to debug. In Section 4.1, we show the effects of under-specifying different operations in pre-processing on image classification models.

The current practice of publishing models also causes a few challenges which must be addressed by a fair and scalable evaluation platform. First, any two ad hoc scripts do not adhere to a consistent evaluation API. The lack of a consistent API makes it difficult to evaluate models in parallel and, in turn, slows down the ability to quickly compare models across different HW/SW stacks. Second, ad

---

*A video demo of web UI is at https://drive.google.com/open?id=1LOXZ7hs_cy-i0-DVU-5FfHwdCd-1c53z.

hoc scripts tend to not clearly demarcate the stages of the model evaluation pipeline. This makes it hard to introspect and debug the evaluation. Furthermore, since an apple-to-apple comparison between models requires a fixed HW/SW stack, it is difficult to perform honest comparison between two shared models without modifying some ad hoc scripts. MLModelScope addresses these challenges through the careful design of a model evaluation specification and a distributed runtime as described in Section 3.

## 3 MLMODELSCOPE DESIGN

We propose MLModelScope, an open-source, framework/hardware agnostic, extensible and customizable distributed system design to perform model evaluation and benchmarking at scale. MLModelScope consists of a model evaluation specification and a distributed runtime.

### 3.1 MODEL EVALUATION MANIFEST

All models in MLModelScope are described using a model specification (called *manifest*). The manifest is a text file and includes the information needed to run a model. The manifest specifies information such as the model pre- and post-processing steps, its model sources (graph and weight), and its software stack. The hardware details are not present in the manifest, but are user-provided options when performing the evaluation.

```
1  name: Inception-v3 # model name
2  version: 1.0.0 # semantic version of model
3  task: classification # model modality
4  license: MIT # model license
5  description: ...
6  framework: # framework information
7    name: TensorFlow
8    version: ^1.x # framework version constraint
9  container: # containers used for architecture
10   arm64: mlms/tensorflow:1-13-0_arm64-cpu
11   amd64:
12     cpu: mlms/tensorflow:1-13-0_amd64-cpu
13     gpu: mlms/tensorflow:1-13-0_amd64-gpu
14   ppc64le:
15     cpu: mlms/tensorflow:1-13-0_ppc64le-cpu
16     gpu: mlms/tensorflow:1-13-0_ppc64le-gpu
17 envvars:
18   - TF_ENABLE_WINOGRAD_NONFUSED: 0
19 inputs: # model inputs
20   - type: image  # first input modality
21     layer_name: data
22     element_type: float32
23 pre-processing: |
24   def pre_processing(env, inputs):
25     ... #e.g. import opencv as cv
26     return preproc_inputs
27 outputs: # model outputs
28   - type: probability # output modality
29     layer_name: prob
30     element_type: float32
31 post-processing: |
32   def post_processing(env, inputs):
33     ... # e.g. os.exec("Rscript ~/postproc.r")
34     return postproc_inputs
35 source: # model source
36   graph_path: https://.../inception_v3.pb
37 training_dataset:  # dataset used for training
38   name: ILSVRC 2012
39   version: 1.0.0
```

Listing 1: Example evaluation manifest.

By decoupling the hardware specification from the manifest, a manifest can work across hardware.

An example manifest is shown in Listing 1 and contains model name, version, and type of task (Lines 1–3); framework name and version constraints (Lines 6–8); containers to use for evaluation (Lines 9–16); model inputs (Lines 19–22); pre-processing function (Lines 23–26); model outputs (Lines 27–30); post-processing function (Lines 31–34); model resources (Lines 35–36); and other metadata attributes (Lines 37–39). The key components of the manifest are:

**Software Stack**−MLModelScope uses docker containers to maintain the software stacks. MLModelScope provides ready-made containers for all popular frameworks, but users can use any container hosted on Docker Hub. Multiple containers can be specified within the manifest. The container used for evaluation is dependent on the executing hardware and whether the system has a GPU or not.

**Model Source**−Model source contains links to the model graph (the `graph_path` field) and weights (the `weights_path` field). For frameworks which have one file to represent the graph and its weights, the weights field is omitted from the manifest. The model can be stored in the cloud, downloaded on demand, and is cached to the local file system.

**Versioning**−Models, frameworks, and datasets are all versioned within MLModelScope using a semantic versioning (Preston-Werner, 2019) scheme. The versioning of frameworks and datasets supports constraints, such as `^1.x` (Listing 1, Line 8). This tells MLModelScope that the model works on any TensorFlow v1 framework.

**Pre-/Post-Processing Functions**−To perform input pre-processing and output post-processing, MLModelScope allows arbitrary Python functions to be placed within the manifest file. The pre- and post-processing functions have the signature `def fun(env, data)` where `env` contains metadata of the evaluation request and `data` is a `PyObject` representation of the user request for pre-processing and the model's output for post-processing. Internally MLModelScope executes the Python code within a Python sub-interpreter (Python Subinterpreter) in the launched container. To reduce data copy overhead parameters are passed by reference to the processing functions. The pre- and post-processing functions are flexible; i.e. users may import external Python modules or invoke external scripts. By allowing arbi-

```
1  type: image   # input modality
2  layer_name: data
3  element_type: float32
4  steps: # pre-processing steps
5    decode:
6      element_type: int8
7      data_layout: NHWC
8      color_layout: RGB
9    crop:
10     method: center
11     percentage: 87.5
12   resize:
13     dimensions: [3, 299, 299]
14     method: bilinear
15     keep_aspect_ratio: true
16   mean: [127.5, 127.5, 127.5]
17   rescale: 127.5
```

Listing 2: MLModelScope's evaluation manifest for Inception-v3.

trary pre- and post-processing function executions, MLModelScope works with existing processing codes and is capable of supporting arbitrary modalities.

**Built-in Pre-/Post-Processing Functions**−As vision models are widely used and their pre- and post-processing operations are less diverse, MLModelScope allows for common pre-processing image operations (e.g. image decoding, resizing, and normalization) and post-processing operations (e.g. topK, IOU, mAP, etc.) to be used within the manifest without writing code. Internally, MLModelScope invokes built-in pre- and post-processing code to perform these operations. Listing 2 can be placed within the inputs block (Lines 19–22 in Listing 1) as the pre-processing steps for Inception-v3. The steps are executed in the order that is specified, since, as we show in Section 4, the order of operations can have a significant impact on achieved model accuracy. Users are not required to use this feature, but using this feature allows users to easily compare pre- or post-processing steps. We use this mechanism during our evaluation in Section 4.

## 3.2 THE MLMODELSCOPE RUNTIME

```
1 // Opens a predictor.
2 rpc ModelLoad(OpenRequest) returns (ModelHandle){}
3 // Close an open predictor.
4 rpc ModelUnload(ModelHandle) returns (CloseResponse){}
5 // Perform model inference on user data.
6 rpc Predict(PredictRequest) returns (PredictionResponse){}
```

Listing 3: MLModelScope's predictor RPC API consists of 3 functions which are specified using Protobuf.

The MLModelScope runtime consumes the model manifest to provision evaluations and perform benchmarking. Users evaluate a model by specifying its name, version, and framework along with the target hardware requirements. The MLModelScope runtime uses these user-provided constraints to query previous evaluations or schedule new ones. The runtime is distributed and is built as a set of extensible and customizable modular components (see Figure 2). Due to space limitations, we only highlight the key components of the runtime (See Appendix for a description of all components):

**Framework Predictors**−At the core of the software stack are the frameworks. To enable uniform evaluation and maximize code reuse, MLModelScope wraps each framework's C++ inference API to provide a uniform interface (called *predictor API*). The predictor API (shown in Listing 3) is minimal and performs model loading, unloading, and inference. So long as a program implements MLModelScope's predictor API, it can be plugged into the system. This means that MLModelScope's design allows for exotic hardware or framework support. For example, some hardware, such as FPGAs and ASICs, do not have a framework per se. These hardware are exposed to MLModelScope through a program which implements the predictor API. The `ModelLoad` API for FPGAs, for example, downloads a bitfile and load it onto the device.

The predictor API is linked against common code to perform container launching, manifest file handling, downloading of required assets, pre- and post-processing function execution, collecting of performance profiles, and publishing of results — we call this bundle an *agent*. These agents can be run on separate machines, can be run in parallel, and are managed by the MLModelScope orchestration layer. Agents can be run on remote systems behind firewalls to allow for model evaluation on remote hardware — this allows hardware providers to give model evaluators access to perform model evaluations without full unrestricted access to the hardware. MLModelScope does not require modifications to a framework and thus pre-compiled binary versions of frameworks (e.g. distributed through Python's pip) or customized versions of a framework work within MLModelScope.

**Manifest and Predictor Registry**−MLModelScope uses a distributed key-value registry (Escriva et al., 2012) to store the model manifests and running agent information. MLModelScope's orchestration layer leverages the registry to facilitate the discovery of models and routing of user requests across the distributed agents using the HW/SW constraints provided by the user. The registry is dynamic — i.e. both model manifests and agents can be added and removed at runtime.

**Profilers and Tracers**−To enable performance debugging, MLModelScope collects system, framework, and model level profiling information. This data is published into a tracing server (OpenTracing; Sigelman et al., 2010) where it gets aggregated and summarized. Through the trace, users get a holistic view of the performance of model evaluation and can identify bottlenecks. To minimize overhead, the profilers are only active when a user enables them as part of the evaluation request.

**Web UI and Command Line Interface**−Users interact with MLModelScope through its web UI or command-line interface by specifying model and hardware constraints. These constraints are used to query the database for previous evaluations or to schedule new ones. Users can integrate MLModelScope within their existing tools or pipelines by using its REST or RPC APIs.

### 3.3 MLMODELSCOPE EVALUATION FLOW

To illustrate the execution flow of a model evaluation, consider a user wanting to run Inception-v3 trained using ILSVRC 2012 on an Intel system with TensorFlow satisfying the "$\geq 1.10.x$ and $\leq 1.13.0$" version constraint. The user specifies these constraints using MLModelScope's UI and invokes the model evaluation. MLModelScope then finds one or more systems which satisfy the user's constraints, sets up the environment, and launches the model evaluation within a container. The results are then published to a database for subsequent analysis.

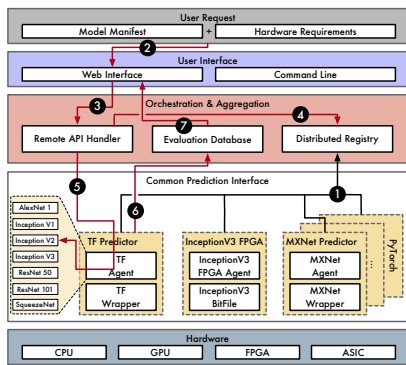

Figure 2: MLModelScope's distributed runtime enables scalable evaluation across models, frameworks, and systems.

Figure 2 shows the evaluation flow of a user's request. ❶ On system startup, each agent publishes the hardware it is running on to the registry. This information is made visible to the MLModelScope orchestration layer. ❷ A user then uses MLModelScope's UI to request an evaluation by specifying the model, framework, and hardware constraints. ❸ An API request is then performed to the remote API handler, which then ❹ queries the registry to find an agent which satisfies the user's constraints. ❺ The request is then forwarded to one (or all) of the agents capable of running the evaluation. The agents then provision the hardware and software environment and run the model. ❻ The agents then collect and publish the results to a centralized evaluation database. ❼ Finally, an evaluation summary is presented to the user.

## 4 EVALUATION

We implemented the MLModelScope design as presented in Section 3 with support for popular frameworks (Caffe, Caffe2, CNTK, MXNet, PyTorch, TensorFlow, TensorRT, and TFLite) and tested it on common hardware (X86, PowerPC, and ARM CPUs as well as GPU and FPGA accelerators). We populated it with over 300 models covering a wide array of inference tasks such as image classification, object detection, segmentation, image enhancement, recommendation, etc. We considered three aspects of MLModelScope for our evaluation: the effects of under-specified pre-processing on model accuracy, model performance across systems, and the ability to introspect model evaluation to identify performance bottlenecks. To demonstrate MLModelScope's functionality, we installed it on multiple Amazon instances and performed the evaluation in parallel using highly cited image classification models.

Unless otherwise noted, all results use TensorFlow `1.13.0-rc2` compiled from source; CUDNN 7.4; GCC 6.4.0; Intel Core i7-7820X CPU with Ubuntu 18.04.1; NVIDIA TITAN V GPU with CUDA Driver 410.72; and CUDA Runtime 10.0.1 (Amazon `p3.2xlarge` Instance).

### 4.1 MODEL PRE-PROCESSING

We use MLModelScope to compare models with different operations in the pre-processing stage. Specifically, we look at the impact of image decoding, cropping, resizing, normalization, and data type conversion on model accuracy. For all the experiments, the post-processing is a common operation which sorts the model's output to get the top $K$ predictions. To perform the experiments, we create variants of the original Inception-v3 (Silberman & Guadarrama, 2018; Szegedy et al., 2016) pre-processing specification (shown in Listing 2). We maintain everything else as constant with the exception to the operation of interest and evaluate the manifests through MLModelScope's web UI.

**Color Layout**−Models are trained with decoded images that are in either RGB or BGR layout. For legacy reasons, OpenCV decodes images in BGR layout by default and, subsequently, both Caffe and Caffe2 use the BGR layout (caffebgr). Other frameworks (such as TensorFlow and PyTorch) use RGB layout. Intuitively, incorrect color layout only misclassifies images which are defined by their colors. Images which are not defined by their colors, however, would be correctly classified. Figure 3 shows the Top 5 classifications for the same image when changing the color layout.

**Data Layout**−Images are represented by: $N$ (batch size), $C$ (channels), $H$ (height), and $W$ (width). Models are trained using data in either NCHW or NHWC form. Figure 4 shows Inception-v3's (trained using NHWC layout) Top1 result using different layouts for the same image.

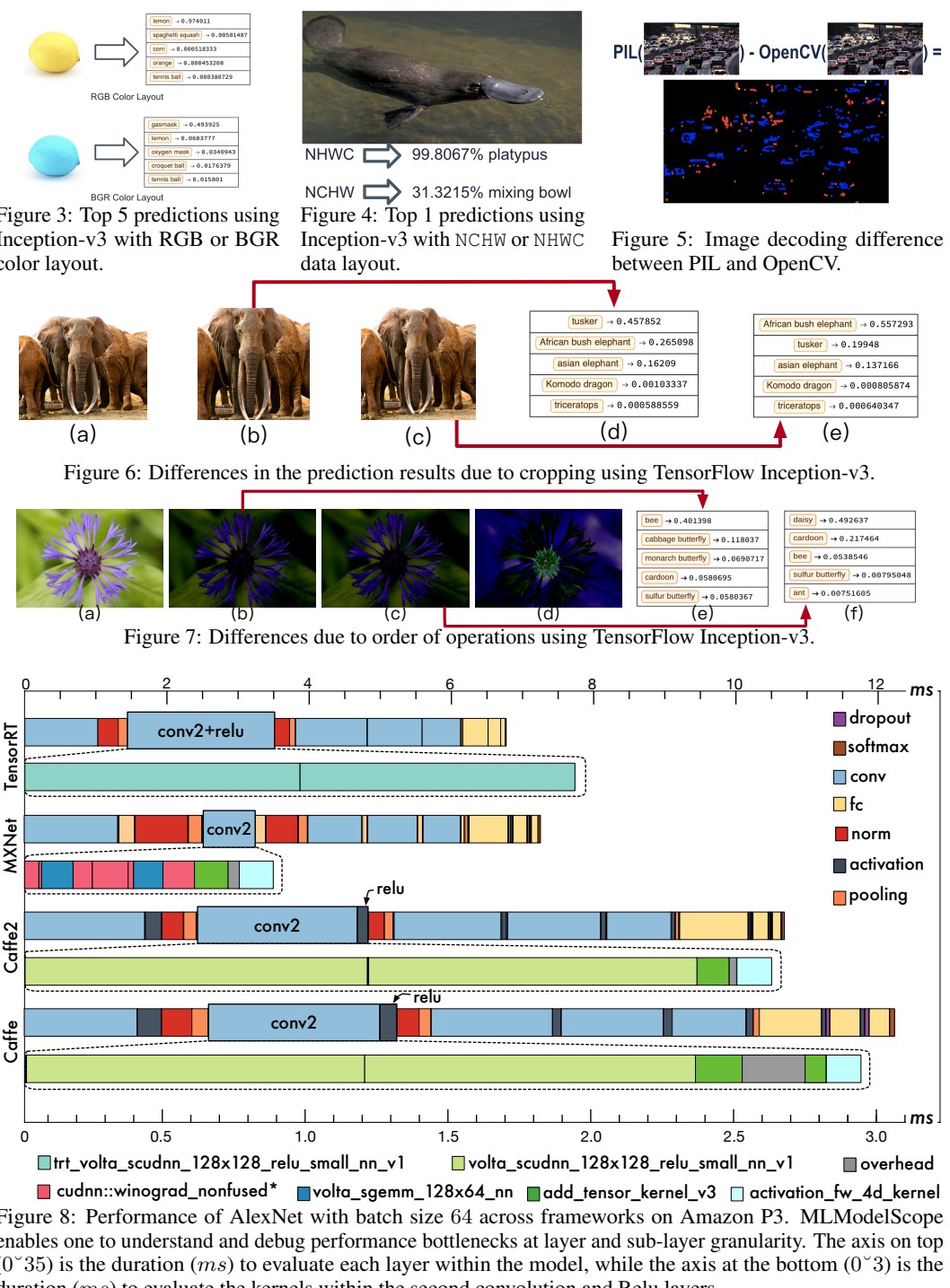

Figure 3: Top 5 predictions using Inception-v3 with RGB or BGR color layout.

Figure 4: Top 1 predictions using Inception-v3 with `NCHW` or `NHWC` data layout.

Figure 5: Image decoding difference between PIL and OpenCV.

Figure 6: Differences in the prediction results due to cropping using TensorFlow Inception-v3.

Figure 7: Differences due to order of operations using TensorFlow Inception-v3.

Figure 8: Performance of AlexNet with batch size $64$ across frameworks on Amazon P3. MLModelScope enables one to understand and debug performance bottlenecks at layer and sub-layer granularity. The axis on top ($0 \check{~} 35$) is the duration ($ms$) to evaluate each layer within the model, while the axis at the bottom ($0 \check{~} 3$) is the duration ($ms$) to evaluate the kernels within the second convolution and Relu layers.

**Decoding and Color Conversion**−It is common to use JPEG as the image data serialization format (with ImageNet being stored as JPEG images). Model developers use library functions such as `opencv.imread`, `PIL.Image.open`, or `tf.io.decode_jpeg` to decode JPEG images. These functions may use different decoding algorithms and color conversion methods. For example, we find the YCrCb to RGB color conversion to not be consistent across the PIL and OpenCV libraries. Figure 5 shows the results[†] of decoding an image using Python's PIL and compares it to decoding

---

[†]To increase the contrast of the image differences on paper, we dilate the image (with radius 2) and adjust its pixel values to cover the range between $0$ and $1$.

| Model Name | Expected | | Color Layout | | Cropping | | Type Conversion | |
|---|---|---|---|---|---|---|---|---|
| | Top1 | Top5 | Top1 | Top5 | Top1 | Top5 | Top1 | Top5 |
| Inception-V3 (Szegedy et al., 2016) | 78.41% | 94.07% | 67.44% | 88.44% | 78.27% | 94.24% | 78.41% | 94.08% |
| MobileNet1.0 (Howard et al., 2017) | 73.27% | 91.30% | 59.22% | 82.95% | 71.26% | 90.17% | 73.27% | 91.29% |
| ResNet50-V1 (He et al., 2016a) | 77.38% | 93.58% | 63.21% | 85.65% | 75.87% | 92.82% | 77.40% | 93.56% |
| ResNet50-V2 (He et al., 2016b) | 77.15% | 93.43% | 63.35% | 85.95% | 75.71% | 92.72% | 77.13% | 93.42% |
| VGG16 (Simonyan & Zisserman, 2014) | 73.23% | 91.31% | 59.17% | 82.77% | 71.71% | 90.61% | 73.24% | 91.33% |
| VGG19 (Simonyan & Zisserman, 2014) | 74.15% | 91.77% | 60.41% | 83.57% | 72.66% | 90.99% | 74.14% | 91.75% |

Table 1: The effects of the pre-processing on the Top 1 and Top 5 accuracy for heavily cited models.

with OpenCV. As shown, edge pixels are not decoded consistently, even though these are critical pixels for inference tasks such as object detection.

**Cropping and Resizing**−Accuracy is sometimes reported for cropped datasets, and this is often overlooked when evaluating a model. For Inception-v3, for example, input images are $87.5\%$ center-cropped and then resized to $299 \times 299$. Figure 6 shows the effect of cropping on accuracy: (a) is the original image; (b) is the result of center cropping the image with $87.5\%$ and then resizing; (c) is the result of just resizing; (d) and (f) shows the top-5 results for images (b) and (c). Intuitively, the effects of cropping are more pronounced for images where the marginal regions are meaningful (e.g. framed paintings).

**Type Conversion and Normalization**−After decoding, the image data is in bytes and is converted to FP32 (assuming an FP32 model). Mathematically, float to byte conversion is $float2byte(x) = 255x$, and byte to float conversion is $byte2float(x) = \frac{x}{255.0}$ and are equivalent. Because of programming semantics, however, the executed behavior of byte to float conversion is $byte2float(x) = \left\lfloor \frac{x}{255.0} \right\rfloor$. The input may also be normalized to have zero mean and unit variance ($\frac{pixel-mean}{stddev}$). We find that the order of operations for type conversion and normalization matters. Figure 7 shows the image processing results using different order of operations for $meanByte = stddevByte = 127.5$ and $meanFloat = stddevFloat = 0.5$ where: (a) is the original image, (b) is the result of reading the image in bytes then normalizing it with both mean and standard deviation in bytes, $byte2float(\frac{imgByte-meanByte}{stddevByte})$, (c) is the result of reading an image in floats then normalizing it with both mean and standard deviation in FP32, $\frac{byte2float(imgByte)-meanFloat}{stddevFloat}$, and (d) is the difference between (b) and (c). The inference results of (b) and (c) are shown in (e) and (f).

Table 1 shows the effects of pre-processing operations [‡] on the top 1 and top 5 accuracy for the entire ImageNet (Deng et al., 2009) validation dataset. The experiments are run in parallel on 4 Amazon `p3.2xlarge` systems. We can see that the accuracy errors due to incorrect pre-processing might be hard to debug, since they might only affect a small subset of the inputs. For example, failure to center-crop the input results in $1.45\% - 7.5\%$ top 1 accuracy difference, and $0.36\% - 4.22\%$ top 5 accuracy difference.

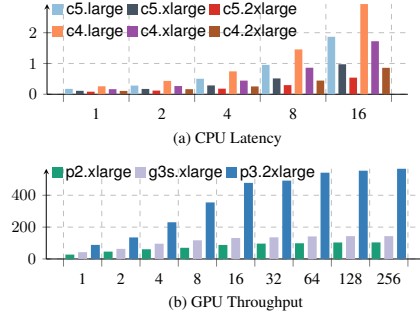

Figure 9: Inference latency of Inception-v3 for (a) CPU and (b) GPU systems. The $x$-axis is the batch size, and the $y$-axis is latency in seconds for (a) and throughput in $images/second$ for (b).

## 4.2 Hardware Evaluation

We use MLModelScope to compare different hardware's achieved latency and throughput while fixing the model and software stack. We launch the same MLModelScope TensorFlow agent on 9 different Amazon EC2 systems recommended for DL (shown in Table 2). These systems are equipped with either GPUs or CPUs. We use MLModelScope's UI to run the evaluations in parallel across all 9 systems, and measure the achieved latency and throughput of the Inception-v3 model as the batch size is varied (shown in Figure 9). Using the measured latency and throughput, along with system pricing information, we calculate the cost/performance as "dollars per million images". We find that GPU instances in general are more cost-efficient than CPU instances for batched inference. We also observe that the `g3s.xlarge` is as cost efficient as the `p3.2xlarge`, because of the high price of the `p3.2xlarge` instance.

## 4.3 Framework Evaluation and Introspection

We use MLModelScope to compare and introspect frameworks' performance by fixing the model and hard-

| Instance | Hardware | $/hr | Cost/Perf. |
|---|---|---|---|
| p2.xlarge | Tesla K80 (Kepler), 12GB | 0.9 | 2.39 |
| g3s.xlarge | Tesla M60 (Maxwell), 8GB | 0.75 | 1.45 |
| p3.2xlarge | Tesla V100-SXM2 (Volta), 16GB | 3.06 | 1.49 |
| c5.large | 2 Intel Platinum 8124M, 4GB | 0.085 | 2.76 |
| c5.xlarge | 4 Intel Platinum 8124M, 8GB | 0.17 | 2.88 |
| c5.2xlarge | 8 Intel Platinum 8124M, 16GB | 0.34 | 3.19 |
| c4.large | 2 Intel Xeon E5-2666 v3, 3.75GB | 0.1 | 5.09 |
| c4.xlarge | 4 Intel Xeon E5-2666 v3, 7.5GB | 0.199 | 5.95 |
| c4.xlarge | 8 Intel Xeon E5-2666 v3, 15GB | 0.398 | 5.94 |

Table 2: Amazon systems used for evaluation.

[‡]We omit from Table 1 the data layout pitfall results, since, as expected, it results in very low accuracy.

ware. For illustration purpose, we show AlexNet, since it has less than 10 layers and fits within the paper. We use MLModelScope's TensorRT, MXNet, Caffe2, and Caffe agents and run them on the Amazon `p3.2xlarge` system. Figure 8 shows AlexNet's latency across frameworks. To understand the performance of each framework, we use MLModelScope's profiler to delve deep and capture each evaluation's layer and library performance information. Through the data, we observe that ML layers across frameworks are implemented differently and dispatched to different library functions. Take the first *conv2* and the following *relu* layers for example. In TensorRT, these two layers are fused and are mapped into two `trt_volta_scudnn_128x128_relu_small_nn_v1` kernels (Oyama et al., 2018) which take $1.95ms$. In Caffe2, however, the layers are not fused and take $2.63ms$. The sub-model profile information helps identify bottlenecks within the model inference. We can see that MLModelScope helps understand the performance across the HW/SW stack which is key to evaluating HW/SW stack choices.

## 5 RELATED WORK

To encourage repeatability in ML/DL research, guidelines (Mitchell et al., 2019; Dodge et al., 2019; Li & Talwalkar, 2019; Lipton & Steinhardt, 2019; Pineau et al., 2018; Reproducibility Checklist) have been developed which authors are advised to follow. These guidelines are checklists of what is required to ease reproducibility and encourage model authors to publish code and write down the HW/SW constraints needed to repeat the evaluation. More often than not, model authors use note-books (Ragan-Kelley et al., 2014), package managers (Fursin et al., 2018a;b) or containers (Kurtzer et al., 2017; Godlove, 2019) to publish their code or specify the SW requirements. These SW requirements are accompanied with a description of the usage, required HW stack, and are published to public repositories (e.g. on GithHub). Through its design, MLModelScope guarantees repeat-able evaluations by codifying the model evaluation through the manifest and user-provided HW constraints.

Both industry and academia have developed consortiums to build benchmark suites that evaluate widely used models (MLPerf; MLMark; AI-Matrix; Gao et al., 2019; Li et al., 2019). These benchmark suites provide separate (non-uniform) scripts that run each model. Each researcher then uses these scripts to perform evaluations on their target HW/SW stack. MLModelScope's model pipeline specification overlaps with the demarcation used by other benchmark suites (e.g. MLPerf seperates model evaluation into pre-processing, model inference, and post-processing). MLModelScope, as an evaluation platform, can incorporate models from benchmark suites so that they can benefit from the distributed evaluation, profiling, and experiment management capabilities. MLModelScope currently has models from benchmark suites such as MLPerf Inference and Alibaba's AI-Matrix built-in.

To allow for distributed evaluation, existing platforms utilize general distributed fabrics (Burns et al., 2016; Boehm et al., 2016; Hindman et al., 2011) to perform model serving (Kubeflow; Chard et al., 2019; Novella et al., 2018; Pachyderm; Zhou et al., 2019) or experimentation (Tsay et al., 2018; FAI-PEP). MLModelScope differs in that it decouples the specification and provisioning of the model evaluation pipeline from the HW/SW stack to enable repeatable and fair evaluations. Moreover, it allows users to introspect the execution at sub-model granularity. To the best of the author's knowledge, no previous design addresses the confluence of repeatability, fairness, and introspection within scalable model evaluation at the same time.

## 6 CONCLUSION

Everyday, an increasingly complex and diverse DL models as well as hardware/software (HW/SW) solutions are proposed — be it algorithms, frameworks, libraries, compilers, or hardware. Both industry and research are hard-pressed to quickly, thoroughly, consistently, and fairly evaluate these new innovations. This paper proposes MLModelScope, which is a specification along with a distributed runtime design that is scalable, extensible, and easy-to-use. Through MLModelScope, users can perform fair and repeatable comparisons across models, software stacks, and hardware. MLModelScope's careful design of the specification, runtime, and parallel evaluation flow reduces time-to-test for model evaluators. With MLModelScope, we evaluate a set of representative image classification models and present insights into how different pre-processing operations, hardware, and framework selection affect model accuracy and performance.

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

## A SUPPLEMENTARY MATERIAL

MLModelScope is a big system, and we had to selectively choose topics due to the space limitation. This supplementary materials section is used to provide details about MLModelScope that we were unable to cover in the paper's main body. Specifically, we discuss how MLModelScope: (a) incorporates the latest research and production models through model manifests by showing object detection and instance segmentation models. (b) Attracts users by providing a web interface and command line for scalable model evaluation. (c) Is built from a set of modular components which allows it to be easily customized and extended.

### A.1 MLMODELSCOPE MODEL MANIFESTS

Listing 4 shows the manifest of `SSD_MobileNet_v1_COCO`, an objection detection model, for TensorFlow. This model embeds the pre-processing operations in the model graph, and thus requires no normalization, cropping, or resizing. The major difference from a image classification model manifest is the task type (being `object_detection`) and the outputs. There are three output tensors for this model (boxes, probabilities, and classes). These output tensors are processed by MLModelScope to produce a single object detection feature array, which can then be visualized or used to calculate the metrics (e.g. mean average precision).

```
1  name: SSD_MobileNet_v1_COCO  # name of your model
2  version: 1.0 # version information in semantic version format
3  task: object_detection # task type
4  framework:
5    name: TensorFlow # framework name
6    version: 1.12.x # framework version contraint
7  container: # containers used to perform model evaluation
8    amd64:
9      gpu: mlcn/tensorflow:amd64-cpu
10     cpu: mlcn/tensorflow:amd64-gpu
11   ppc64le:
12     cpu: mlcn/tensorflow:ppc64le-gpu
13     gpu: mlcn/tensorflow:ppc64le-gpu
14 description: ...
15 references: # references to papers / websites / etc.. describing the model
16   - ...
17 license: Apache License, Version 2.0 # license of the model
18 inputs: # model inputs
19   - type: image # first input modality
20     element_type: uint8
21     layer_name: image_tensor
22     layout: HWC
23     color_layout: RGB
24 outputs:
25   - type: box
26     element_type: float32
27     layer_name: detection_boxes
28   - type: probability
29     element_type: float32
30     layer_name: detection_scores
31   - type: class
32     element_type: float32
33     layer_name: detection_classes
34     features_url: https://.../labels.txt
35 source:
36   graph_path: https://.../ssd_mobilenet_v1_coco_2018_01_28.pb
37 attributes: # extra model attributes
38   training_dataset: COCO # dataset used to for training
39   manifest_author: ...
```

Listing 4: MLModelScope's model specification for `SSD_MobileNet_v1_COCO` TensorFlow model.

Listing 5 shows the manifest of `Mask_RCNN_ResNet50_v2_Atrous_COCO`, an instance segmentation model, for MXNet. The major difference from the object detection model in Listing 4 is the task type (being `instance_segmentation`) and the outputs. Listing 5 shows four outputs for this model (boxes, probabilities, classes, and masks). These output tensors are processed by MLModelScope to produce a single instance segmentation feature array. Note that unlike TensorFlow, MXNet uses layer indices in place of layer names to get the tensor objects.

```
1  name: Mask_RCNN_ResNet50_v2_Atrous_COCO # name of your model
2  version: 1.0 # version information in semantic version format
3  task: instance_segmentation
4  framework:
5    name: MXNet # framework for the model
6    version: 1.4.x # framework version contraint
7  container: # containers used to perform model evaluation
8    amd64:
9      gpu: mlcn/mxnet:amd64-cpu
10     cpu: mlcn/mxnet:amd64-gpu
11   ppc64le:
12     cpu: mlcn/mxnet:ppc64le-gpu
13     gpu: mlcn/mxnet:ppc64le-gpu
14 description: ...
15 references: # references to papers / websites / etc.. describing the model
```

```
16     - ...
17  license: Apache License, Version 2.0 # license of the model
18  inputs: # model inputs
19     - type: image # first input modality
20       element_type: uint8
21       layout: HWC
22       color_layout: RGB
23  outputs:
24     - type: box
25       element_type: float32
26       layer_name: 0
27     - type: probability
28       element_type: float32
29       layer_name: 1
30     - type: class
31       element_type: float32
32       layer_name: 2
33       features_url: https://.../labels.txt
34     - type: mask
35       element_type: float32
36  source: # specifies model graph and weights sources
37     base_url: http://.../mxnet/Mask_RCNN_ResNet50_v2_Atrous_COCO/
38     graph_path: model-symbol.json
39     weights_path: model-0000.params
40  attributes: # extra model attributes
41     training_dataset: COCO # dataset used to for training
42     manifest_author: ...
```

Listing 5: MLModelScope's model specification for `Mask_RCNN_ResNet50_v2_Atrous_COCO` MXNet model.

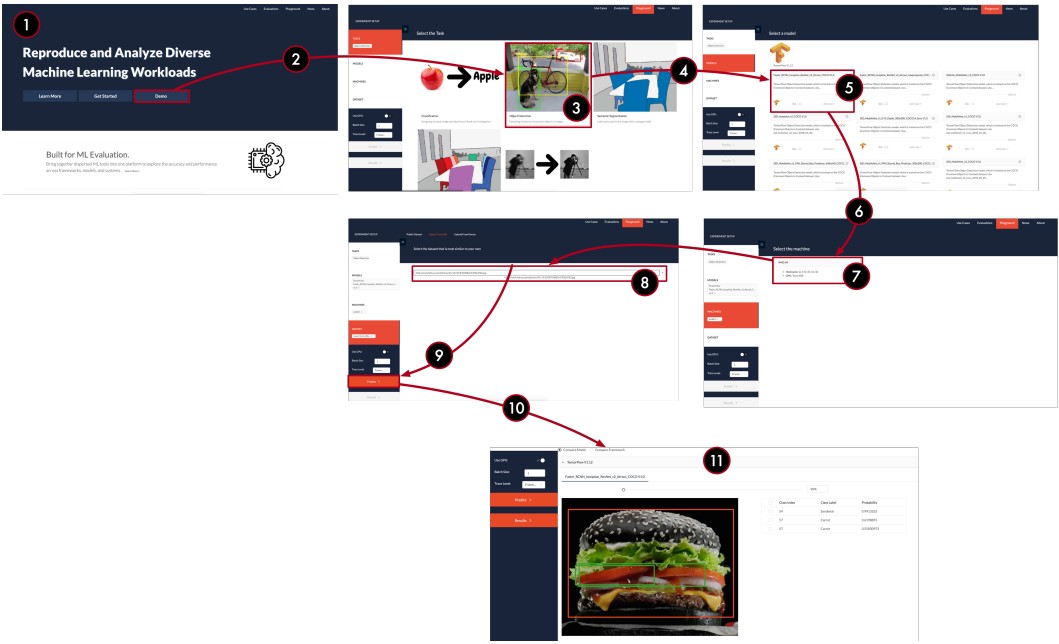

Figure 10: The MLModelScope website provides an intuitive interface to conduct experiments.

## A.2 WEBSITE WORKFLOW

Although MLModelScope provides both command line and library interfaces, we find the website provides an intuitive flow for specifying and running experiments. Figure 10 shows the flow, and a video demonstrating it can be found at https://drive.google.com/open?id=1LOXZ7hs_cy-i0-DVU-5FfHwdCd-1c53z. In figure 10, users first arrive at ❶ MLModelScope's landing page. The landing page contains a description of the project along with links to how to setup and install MLModelScope. Users can try MLModelScope by ❷ clicking the demo button, which then displays ❸ the inference tasks exposed through the website. If a user ❹ selects object detection, then ❺ models that are available for object detection are displayed. A user can then ❼ selects one or more models and ❽ selects one or more systems to run the evaluation on. The input can be specified as a URL, data from disk, or dataset ❽ and once complete the user can perform the evaluation ❾. This ❿ will run the evaluation on the remote system and ⓫ display the evaluation results along with summary of the execution flow.

### A.3 MLModelScope's Runtime Architecture

In this section we describe each component in Figure 11 in detail. The runtime is designed to be extensible and customizable.

#### A.3.1 User Interface and API

MLModelScope can be used as an application or as a library. Users interact with MLModelScope application through its website, command line, or its API interface. The website and command line interface allow users to evaluate and profile models without familiarity with the underlying frameworks or profiling tools. Users who wish to integrate MLModelScope within their existing tools or pipelines can use the REST or RPC APIs. They can also compile MLModelScope as a standalone shared library and use it within their C/C++, Python, or Java projects.

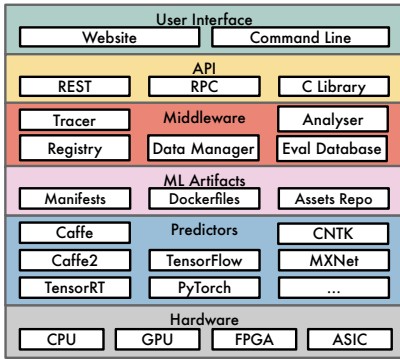

Figure 11: MLModelScope's runtime components.

#### A.3.2 ML Artifacts

As discussed in the main body of the paper, replication of model accuracy and performance results is dependent on: the usage of specific HW/SW stack; the training dataset; and the pre/post-processing steps on the inputs and outputs. MLModelScope specifies these requirements via a model manifest file described in Section 3. The manifest tells MLModelScope the HW/SW stack to instantiate and how to evaluate the model.

**Asset Versioning** — Models, frameworks, and datasets are versioned using a semantic versioning scheme. The MLModelScope middleware layer uses this information for asset management and discovery. To request a model, for example, users specify model, framework, hardware, or dataset constraints. MLModelScope solves the constraint and returns the predictors (systems where the model is deployed) that satisfy the constraint. The model evaluation can then be run on one of (or, at the user request, all) the predictors.

**Docker Containers** — To maintain the SW stack, evaluations are performed within docker containers. To facilitate user introspection of the SW stack, MLModelScope integrates with existing docker tools that allows querying images's SW environment and metadata.

**Pre/Post-Processing Operations** — MLModelScope provides the ability to perform common operations such as resizing, normalization, and scaling without writing code. It also allows users to specify code snippets for pre/post-processing within the manifest file which are run within a Python subsession. MLModelScope is able to support a wide variety of models for different input modalities.

**Evaluation History** — MLModelScope uses the manifest information as keys to store the evaluation results in a database. Users can view historical evaluations through the website or command line using query constraints similar to the ones mentioned above. MLModelScope summarizes and generates plots to aid in comparing the performance across experiment.

#### A.3.3 Framework and Model Predictors

A predictor is a thin abstraction layer that exposes a framework through a common API. A predictor is responsible for evaluating models (using the manifest file) and capturing the results along with the framework's profile information.A predictor publishes its HW/SW stack information to MLModelScope's registry at startup, can have multiple instantiations across the system, and is managed by MLModelScope's middleware.

#### A.3.4 Middleware

The middleware layer is composed of services and utilities for orchestrating, provisioning, aggregating, and monitoring the execution of predictors — acting as a conduit between the user-facing APIs and the internals of the system.

**Manifest and Predictor Registry** — MLModelScope uses distributed key-value database to store the registered model manifests and running predictors. MLModelScope leverages the registry to facilitate discovery of models, load balancing request across predictors, and to solve user constraint for selecting the predictor (using HW/SW stack information registered). The registry is dynamic — both model manifests and predictors can be added or deleted at runtime throughout the lifetime of the application.

**Data Manager** — MLModelScope data manager is responsible for downloading the assets (dataset and models) required by the model's manifest file. Assets can be hosted within MLModelScope's assets repository, or hosted externally. For example, in Listing 1 ( Lines 35–36) the manifest uses a model that's stored within the MLModelScope repository, the data manager downloads this model on demand during evaluation.

Within MLModelScope's repository, datasets are stored in an efficient data format and are placed near compute on demand. The dataset manager exposes a consistent API to get values and iterate through the dataset.

**Tracer** — The MLModelScope tracer is middleware that captures the stages of the model evaluation, leverages the predictor's framework profiling capability, and interacts with hardware and system level profiling libraries to capture fine grained metrics. The profiles do no need to reflect the wall clock time, for example, users may integrate a system simulator and publish the simulated time rather than wall-clock time.

MLModelScope publishes the tracing results asynchronously to a distributed server — allowing users to view a single end-to-end time line containing the pipeline traces. Users can view the entire end-to-end time line and can "zoom" into specific component (shown in Figure 1) and traverse the profile at different abstraction levels. To reduce trace overhead, users control the granularity (AI component, framework, library, or hardware) of the traces captured.

MLModelScope leverages off-the-shelf tools to enable whole AI pipeline tracing. To enable the AI pipeline tracing, users inject a reference to their tracer as part the model inference API request to MLModelScope. MLModelScope then propagates its profiles to the injected application tracer instead of the MLModelScope tracer — placing them within the application time line. This allows MLModelScope to integrate with existing application time lines and allows traces to span API requests.

### A.3.5 FRAMEWORKS

At time of writing, MLModelScope has built-in support for Caffe, Caffe2, CNTK, MXNet, PyTorch, Tensorflow, TFLite, and TensorRT. MLModelScope uses "vanilla" unmodified versions of the frameworks and uses facilities within the framework to enable layer-level profiling — this allows MLModelScope to work with binary versions of the frameworks (version distributed through Python's pip, for example) and support customized or different versions of the framework with no code modifications. To avoid overhead introduced by scripting languages, MLModelScope's supported frameworks use the frameworks' C-level API directly — consequently the evaluation profile is as close to the hardware as possible.

### A.3.6 HARDWARE

MLModelScope has been tested on X86, PowerPC, and ARM CPUs as well as NVIDIA's Kepler, Maxwell, Pascal, and Volta GPUs. It can leverage NVIDIA Volta's TensorCores, and can also perform inference with models deployed on FPGAs. During evaluation, users can select hardware constraints such as: whether to run on CPU or GPU, type of architecture, type of interconnect, and minimum memory requirements — which MLModelScope considers when selecting a system.

### A.4 MLMODELSCOPE SOURCE CODE

This project is open-source and the code spans multiple ($> 15$) repositories on GitHub. Thus it is difficult to anonymize for the blind review process. We are happy to share the links to the source code with the PC members. The links will be included in the paper after the blind review process.

