# OpenReview forum: "MLModelScope: A Distributed Platform for ML Model Evaluation and Benchmarking at Scale"
_ICLR.cc/2020/Conference — Reject_

### Official Review · AnonReviewer1 · 2019-10-23
**Official Blind Review #1**

**Rating:** 6

**Review:**

The paper presents a unified approach to specify, evaluate and benchmark different ML methods.
With the main goal of enforcing repeatability and faireness when testing different methods, authors propose
an open source runtime on which 1) specify the model, 2) describe the workflow and 3) evaluate the benchmark
of several ML algorithms and frameworks.
The core of the work is the definition of the so-called "model evaluation manifest" which consists of a formatted
collection of descriptive information where both hardware/software and framework versions are specified, along with
the set of tasks to be carried on as well as the data sources to test the methods against.
Once the manifest has been created, the desired hw/sw configuration is deployed on Amazon and the specified models are benchmarked.
This benchmarking offers several insights on the evaluation of a given ML model, by stressing out the importance of aspects that can severely bias  the final outcome of the model (e.g., pre-processing tasks, different hardware configurations or normalization of the data).
To describe the workflow, authors use an image classifier on a given hardware as a running example, and play with different  preprocessing methods to measure their impact on the final accuracy of the model.

Some details are not well specified/clear in the work:
1) Data exploitation. There is the possibility of testing different methods on own datasets. Given that the deployment is run on Amazon instances, what are the requirements (e.g., data must be on S3 and so on).
2) The manifest can be injected with python scripts that, running in a container, perform the desired operations (preprocessing). It is stated that "parameters are passed by reference". So if you pass a "mutable" object ("env", I guess) you need to bind it to the outher scope. How this is accomplished? (globals?)
Instead, if you pass an "immutable" object ("data", I guess), you cannot rebind the outer reference nor mutate the object. So, what's the meaning of "passing by reference"?
3) Privacy and anonymity. When performing debugging, system, framework and model level profiling information are collected on a tracing server. Is this server  part of the platform?

**Experience Assessment:**

I have read many papers in this area.

**Review Assessment: Checking Correctness Of Derivations And Theory:**

I assessed the sensibility of the derivations and theory.

**Review Assessment: Checking Correctness Of Experiments:**

I assessed the sensibility of the experiments.

**Review Assessment: Thoroughness In Paper Reading:**

I read the paper at least twice and used my best judgement in assessing the paper.

---

> ### Author Response · Authors · 2019-11-11
> **Author Response to Official Blind Review #1**
>
> 1. Data exploitation
> MLModelScope can be deployed on any hardware system and is not tied to Amazon instances or services. In fact, the authors regularly run MLModelScope on local systems. We chose AWS instances for evaluation for two reasons: (1) it is easy to scale out the experimentation on multiple systems and use Amazon Machine Image (AMI) to guarantee the same software environment on the systems; (2) AWS systems are accessible to others who can repeat the claims made in this paper.
>
> The data manager (a common component across predictors as described in Appendix 3.4) manages the datasets. The data manager provides an abstract interface to iterate over HDFS, MXNet's RecordIO, TensorFlow's TFRecord, or plain raw files. Evaluation datasets are embedded into the data manager as a simple lookup table. The entry for each dataset in the lookup table include the dataset's name, version, URL, and format. MLModelScope currently includes popular datasets such as ImageNet, MNIST, MSCoco which are stored on S3, Zendo, etc. Since datasets are built-in, they are identified by their name and version as shown in Listing 1 line 36. The data manager downloads the dataset from the URL given its name and version information. To add a new dataset for evaluation,  a user just needs to add the dataset's name, version, URL, and format as an entry in the lookup table.
>
> In summary, MLModelScope is not tied to AWS, the model assets and datasets can reside on any URL, and users can evaluate models on their own datasets.
>
> 2. Pass by Reference
> MLModelScope is written in GoLang and not Python. We choose GoLang since it is commonly used to build distributed applications (e.g. Kubernetes is written in GoLang) and has speed comparable to C/C++. Since MLModelScope is written in GoLang, to support pre- and post-processing steps written in Python, there needs to be a translation layer between the data resident within the MLModelScope runtime and the PyObjects (passed to the Python subinterpreter). Not sharing the data between the two is detrimental to performance, and so we developed a shallow copy mechanism to translate between the two languages.
>
> To perform the shallow copy, the MLModelScope runtime packs its owned data as a PyObject data structure. Since the input data can be a container, there are a set of rules that are followed when packing the data into the PyObject: (1) for atomic scalars a PyObject is created, (2) for homogenous rectangular data list (i.e. Tensors), a NumPy object is created where the underlying data is shared with the MLModelScope runtime, (3) for non-homogeneous data list, each element is packed and a PyObject is created from the packed list, (4) for hash tables (dictionaries), the values of the hash table are packed and PyObjects are created for the keys of the dictionary. A PyObject is created for the packed dictionary values and the PyObject keys. A sketch of the algorithm to perform shallow copy from GoLang (goData) into a PyObject is shown below:
>
> def pack_data(goData) -> PyObject:
>      if isinstance(goData, (int, float, double, …)): # all atomic scalars
>         return mk_pyobj(goData)
>      if isinstance(goData, (list, tuple)) and can_create_numpy(data): # is numpy
>         return mk_numpy_pyobj(goData) # share the underlying data
>      if isinstance(goData, (list, tuple)): # is list
>         return mk_pyobj([pack_data(elem) for elem in goData])
>      if isinstance(goData, dict): # is dictionary
>         return mk_pyobj({mk_pyobj(k): pack_data(v) for k, v in goData.items()})
>      throw "unhandled type {}".format(typeof(goData))
>
> The key here is that mk_numpy_pyobj does not clone the underlying data; rather it uses the same goData owned by the MLModelScope runtime. The underlying array data would therefore be shared between the MLModelScope runtime and the Python subinterpreter.
>
> As stated in the paper, the env field contains metadata and therefore is passed by value (i.e. serialized to a PyObject) to the Python subinterpreter. It is only the second parameter (data) that is shared between the MLModelScope runtime and the Python sub-interpreter.
>
> Since this is an implementation detail, rather than part of the design, we only briefly mentioned “parameters are passed by reference” as an optimization within the implementation. We will clarify this in the paper.
>
> 3. Privacy and anonymity
> As described in Appendix 3.4, the tracing server is part of the MLModelScope platform and we use an open source implementation (Jaeger). The tracing server can run on any system and its IP is made known to MLModelScope through a configuration file. MLModelScope tracers publish the captured profiling information (trace) to the tracing server and the trace ID is associated with the evaluation and the user. The MLModelScope UI supports a concept of a “user” and only evaluations belonging to the user are viewable to the user.

---

### Official Review · AnonReviewer2 · 2019-10-26
**Official Blind Review #2**

**Rating:** 6

**Review:**

This paper studies the question of model evaluation and reproducibility in machine learning research, specifically deep learning research, and designs and tests an extensive system for evaluating and comparing models. Users specify their evaluation parameters through a text file, and this is used by the runtime, which can be interfaced through a UI or the command line, in order to carry out the evaluation. Several experiments shed interesting light on various aspects of model, framework, and hardware performance.

Hypothetically, if I were to design a new model and wish to evaluate its performance relative to existing SotA models, I would potentially use this system to run all of the models, including my own. That would mean that I need to "upload" or otherwise integrate my model into this system, and it was unclear from my reading of the paper how easy such a process would be. Similarly, I would wish to maintain similar training and evaluation conditions for my model, e.g., the same pre and post-processing, and that would involve "extracting" those steps from the system for use during training. I would also like to understand whether or not this is feasible and easy given the system's design.

In section 3.1, the authors write "The hardware details are not present in the manifest, but are user-provided options when performing the evaluation." An example of how this operates would be useful in the paper.

As far as experiments go, I'm not sure what the main takeaway is from section 4.1. To me, the takeaway that pre-processing is important and existing models are sensitive to pre-processing is not a new finding. The results from Table 1 could certainly be obtained without the use of the proposed system, and though there would be some scaffolding involved, I don't think that the coding would not be particularly difficult. Is the takeaway that the proposed system makes it easier or faster to evaluate the effects of different types of pre-processing? Wouldn't this be most interesting at training time?

I find the experiments in sections 4.2 and 4.3 interesting. In section 4.2, I'm not sure if figure 9 includes enough information or description to conclude that "GPU instances in general are more cost-efficient than CPU instances for batched inference", and some more detail here would be useful.

Generally, I believe that the work is well-motivated and timely, the authors seem to have done a good job in citing related work (though admittedly I don't know much about this area), and the results are supportive of the claims of the system's usefulness.

**Experience Assessment:**

I do not know much about this area.

**Review Assessment: Checking Correctness Of Derivations And Theory:**

N/A

**Review Assessment: Checking Correctness Of Experiments:**

I assessed the sensibility of the experiments.

**Review Assessment: Thoroughness In Paper Reading:**

I read the paper at least twice and used my best judgement in assessing the paper.

---

> ### Author Response · Authors · 2019-11-11
> **Author Response to Official Blind Review #2**
>
> 1. Specify a model
> As shown in Figure 2 and discussed in Section 3.1, a user inputs the model manifest as part of the evaluation request. The model manifest, together with the user-specified hardware requirements form the constraints to the MLModelScope runtime. The runtime uses these constraints to resolve the framework predictor(s) that are capable of performing the evaluation (circle 4). For ease of use, we boot-strapped MLModelScope with over 300 models from common frameworks by embedding the model manifests within the corresponding framework predictor. At the startup phase, the running predictors register the built-in models in the registry (circle 1). To use the built-in models, a user just specifies the model and framework name and version in place of the model manifest. In summary, models are defined through the model manifest file and no coding is required to add models. Users use the website or command line interfaces to either select built-in models or input full model manifests.
>
> 2. Extracting pre- and post-processing
> We do require extracting the pre- and post-processing steps and this is key to enable fair comparison across models. To perform a fair comparison of two models requires demarcating the pre-processing, prediction, and post-processing steps. In fact, it is common to have pre- and post-processing code as functions, since they can be used across models performing the same ML task and for both inference and training. Examples are found in TensorFlow's model zoo or MXNet's Gluon library. Thus, we do not think it is much of an ask to extract the pre- and post-processing code to ensure a fair comparison. With that said, the pre- and post-processing in MLModelScope allow for arbitrary code. Thus, the reviewer's scenario would be feasible using their existing processing code.
>
> In practice, when benchmarking model inference, the prediction step is what is compared (e.g. in popular ML benchmark suites such as MLPerf and AI-Matrix). For example, MLPerf inference separates the prediction from both the pre-processing and post-processing steps. Unlike these benchmarking suites, which manually write custom scripts that separate these steps, MLModelScope performs the demarcation by design. To the authors' knowledge, we are the first to design a specification which provides a systematic way to enforce fair evaluation.
>
> 3. Specification of hardware details
> As shown in Figure 2, during startup all agents self-register by populating the registry with their hardware information, available built-in models, and their software stacks (circle 1). To initiate an evaluation, a user inputs the model manifest and hardware requirements (circles 2&3).  The MLModelScope runtime then resolves these constraints, queries the registry (circle 4), and then selects agent(s) to dispatch the evaluation request to (circle 5). The model manifest and hardware specification are designed to be decoupled so that one can easily evaluate different combinations (e.g. evaluating a model manifest across different hardware systems).
>
> 4. Section 4.1
> As reviewer #2 recognized, Section 4.1 is used as a case study to show that, through MLModelScope, one can easily explore different pre-processing methods and measure their impact on the accuracy of the model. MLModelScope makes this process easy, as the experiments can be initiated with a few clicks using web UI or a single invocation of the command line interface. The experiments are then run in parallel, the results are collected and stored in a database. One can no doubt perform the same evaluation manually, but MLModelScope streamlines and simplifies the model evaluation process.
>
> The community acknowledge that reproducibility is a “pain point” and not all people are aware of all the pitfalls presented. And, while most people expect that pre-processing has an effect on model accuracy, this case study quantifies the effects.
>
> While we agree with the reviewer that a platform similar to MLModelScope with in-depth evaluation is needed for training, such a platform is missing for both inference and training. MLModelScope, as presented in this paper, is the first step to address the inference issues,. We leave the extension to training as our future work.
>
> 5. Section 4.2
> The observation is made based on the cost/performance (Cost/Perf) information listed in Table 2. As stated in the paper, Cost/Perf are computed using the measured performance in Figure 9(maximum throughput at the optimal batch size) and the AWS pricing($/hr) in Table 2. The unit is dollars per million images. This observation only applies to the model and AWS instances used in the experimentation. The authors do expect the cost-effectiveness of GPU vs. CPU depends on the model, GPU, CPU, software stack, and the instance pricing and a recent study discusses this in cloud serving scenario [1].
>
> [1]  “MArk: Exploiting Cloud Services for Cost-Effective, SLO-Aware Machine Learning Inference Serving”, USENIX ACT2019.

---

### Official Review · AnonReviewer4 · 2019-11-14
**Official Blind Review #3**

**Rating:** 1

**Review:**

* Note: I highlight I did not assess the model design, which is the main contribution of the paper, and  did not know the background of prior work of system design to really assess the novelty of the work, my score is solely based on the experiments.
 I am an expert in machine learning/computer vision, so I could assess the experiments in terms of their validity and relevance from the machine learning/vision perspective, however, I may not be the best person to evaluate the design choices of the system. Therefore I choose the low experience for my background.

* Paper summary: The paper proposes a framework to evaluate machine learning models in a hardware-agnostic way.
To evaluate the models using this framework, the user needs to specify the pre-processing, inference, and post-processing steps and the required software/hardware stack. The authors argue that this is important to consider the  HW/SW stack to allow a fair evaluation and reproducibility. Models are specified using a model specification called manifest.

* The authors assume that SW/HW stack change the results of deep learning models a lot, and this is the main assumption in this work, however, normally in practice HW/SW stack wont change the results.

* I found the experiments either not related to the point of the paper or being very trivial not helping to backing up the arguments of the paper.


* In section 4.1, the authors consider different preprocessing operations and study their impact on the model performance, however, the fact that preprocessing impact the results is trivial in machine learning. In the same section, color layout and data layout, cropping and resizing, where the authors discussed about for instance how changing the data representation from NCHW or NHWC change the results, this is also trivial, because if you change the dimensions, you need to also change the model in a way that handles this change of dimension, therefore, this is clear that the results will change accordingly as well. Such experiments does not back up the main argument of the paper, which argues for fair evaluation between neural models, nor provides informative information to the reader.

On section 4.1, the experiment of type conversion and normalization, again this is mathematically clear that the order would change the results,  let's call imgByte=x, then by substituting given
values for mean and standard evaluation, equation (b) is simplified to (b) = (x-127.5)/((127.5)*(255))
however simplification of (c) results in (x/255-0.5)/0.5 = (x-0.5*255)/(0.5*255)=(x-127.5)/(0.5*255)
the dominator of (b) and (c) are not equal, therefore, this is trivial that the results of these two
the expression would not be the same. The author posed it as a new finding, but this is trivial that mathematically
these two equations would not be equal. Again, this experiment does not add any value to the paper.

In section 4.2, in Figure 9, the authors show a plot of the CPU latency for different batch sizes and instances,
together with GPU throughput for different batch sizes, i.e., images/seconds. The authors show latency for CPU
instances, versus throughput for GPU instance, since these two measures are not shown for both instances, this is
not supported from the text, how actually authors compare this two instance and draw the conclusion that which instance is more efficient since there is no value shown for CPU throughput. Apart from that, I don't see how this section and determining if GPU or CPU instances of  Amazon compute cloud is more cost-efficient is related to the point of this paper which is on reproducibility. Also please have a look at Amazon webpage:
https://docs.aws.amazon.com/dlami/latest/devguide/gpu.html
Here, they explicitly mention that "A GPU instance is recommended for most deep learning purposes. Training new models will be faster on a GPU instance than a CPU instance. You can scale sub-linearly when you have multi-GPU instances or if you use distributed training across many instances with GPUs.",  so having this experiment again neither back up the arguments in the paper, nor add value to the paper.

* The major issue with this submission is that the experiments are not related to the arguments of the paper, and are not conveying any message towards backing up the arguments of the paper.

* Another crucial problem is that to allow a fair comparison especially in neural models, as shown in several studies(see [1] as a sample), this is important to account for random seeds and study how it impacts the model performance, to allow a fair evaluation of the models this is important to consider this factor, fair evaluation of models is argued to be the main point of this paper, however, the authors does not consider this factor in the paper, nor study it in the experiments.

[1] Sentence-BERT: Sentence Embeddings using Siamese BERT-Networks, Nils Reimers and Iryna Gurevych





**Experience Assessment:**

I have read many papers in this area.

**Review Assessment: Checking Correctness Of Derivations And Theory:**

N/A

**Review Assessment: Checking Correctness Of Experiments:**

I carefully checked the experiments.

**Review Assessment: Thoroughness In Paper Reading:**

I read the paper thoroughly.

---

> ### Author Response · Authors · 2019-11-14
> **Official Blind Review #3**
>
> Thank you for the detailed reviews. We would like to clarify the contribution of the paper and the purpose of the evaluation section before we address the questions about Section 4.1 and 4.2.
>
> This paper is about software and tooling for scalable ML evaluation and benchmarking, thus the main contribution comes from the design. The challenge we addressed through MLModelScope is to design a scalable ML benchmarking platform while guaranteeing repeatable and fair evaluation. As Reviewer#1 summarized: “The paper presents a unified approach to specify, evaluate and benchmark different ML methods”, and we believe we are the first to propose such a design.
>
> The goal of MLModelScope is to boost the productivity of users to evaluate and benchmark ML models. The evaluation section serves as case studies to showcase the types of experiments or comparisons (4.1 is about model, 4.2 is about hardware, and 4.3 is about framework) that can be performed through MLModelScope. This paper claims the novelty of the design but never claims novelty of the results of the experiments.
>
> 1. Section 4.1
> The authors agree with the reviewer's statement: "the fact that preprocessing impact the results is trivial". Still, the community acknowledges that reproducibility is a “pain point” and not all people are aware of all the pitfalls presented (See citations in the introduction). And, while most people expect that pre-processing has an effect on model accuracy, this case study quantifies the effects. Therefore, Section 4.1 is presented as a case study. We use MLModelScope to enumerates some of the common pre-processing pitfalls and quantify their effects on the model's accuracy.
>
> Type conversion and normalization --- First, there is a typo in the paper where it says byte2float(x) = floor(x/255.0) that should have been float2byte(x) = floor(x*255). Furthermore, as the reviewer surmised, the formula in (b) is not correct and the expression should have been byte2float(imgByte-meanByte)/byte2float(stddevByte). One thing the paper does not clearly state is that when performing the pre-processing in the byte domain the values of meanByte and meanStddev are implicitly represented as 127 (which is  float2byte(0.5)) and not 127.5. In summary, x is image pixel, (b) = byte2float(x-127)/byte2float(127) = (x-127)/127.0, and (c) = (byte2float(x)-0.5)/0.5 = (x/255.0-0.5)/0.5 = (x-127.5)/127.5. Regardless of the typo, the experiments hold (e.g. there would be a significant visible difference in Fig7(b,c)). We provide a sketch implementation at https://bit.ly/33NP2TQ that validates our claim. We will clarify this in the paper.
>
> 2. Section 4.2
> The purpose of Section 4.2 is to serve as a case study to show through MLModelScope, the user can easily perform evaluations across hardware at scale and use it (along with pricing information) to select the most cost-effective hardware for their workload. The Amazon reference is a rough guide for training, NOT model deployment (inference). The cost-effectiveness of GPU vs. CPU depends on the model, GPU, CPU, software stack, and instance pricing.
>
> CPU is latency-oriented hardware and GPU is throughput-oriented hardware, thus Figure 9 shows the latency for CPU and throughput for GPU so that readers can easily look up numbers or do a sanity check. Throughput (images/second) is defined as the inverse of latency (second), i.e. throughput = 1/latency. Thus, the performance information for both CPUs and GPUs (as presented in Figure 9) is complete. As described in response to Reviewer #2 Question 5, the cost efficiency observation follows from the cost/performance (Cost/Perf) information listed in Table 2. As stated in the paper, the Cost/Perf is computed using the measured performance in Figure 9 (maximum throughput at the optimal batch size) and the AWS pricing($/hr) in Table 2. The unit is dollars per million images. We will clarify this in the paper.
>
>
> As this paper is about the ML software and tooling, we sincerely ask the reviewer to assess the MLModelScope design. We’d greatly appreciate it if the reviewer can reconsider the “Experience Assessment” since the reviewer did not evaluate the system's design, which is the main contribution of the paper. Note that Reviewer#1 indicates the same level of experience assessment.

---

> > ### Comment · AnonReviewer4 · 2019-11-14
> > **Models Novelty - Experiments**
> >
> > * regarding the novelty of the paper on scalability, I would like to point out that being able to run deep learning models at scale,  is well-provided in most of the deep learning frameworks, as an example, here is extensive PyTorch documents on writing distributed models: https://pytorch.org/docs/stable/distributed.html
> > Tensorflow also well supports this, and writing scalable codes is generally straight forward, could possibly get implemented in a few lines of codes, at least in Pytorch.
> >
> > * The major issue regarding the experiments is that paper argues that the proposed model allows "fair and reproducible" benchmarking, however, the experiments are not at all backing this up, and I found them not being related to the point of the paper. For instance, the authors argue that the order of preprocessing stages can change the results, or preprocessing impacts the final results, but this is really trivial in machine learning. Also, on open-source codes, the order of preprocessing stages are specified, I am not seeing these experiments as informative nor related.
> >
> > * About the experiments on choosing CPU/GPU instances of amazon to compute cloud, this is as I mentioned in my first comment pointed out on Amazon's webpage, still, I cannot see how this is related to reproducibility and allowing fair benchmarking of deep learning models.
> >
> > * one major issue is that random seeds have been shown to substantially impact the performance of neural models, see [1] for an example, the proposed model of evaluation in the paper, does not account for this crucial factor, I would expect that in the proposed framework consider running the models for different seeds and study how this is impacting the results. However, this is not considered.
> >
> > [1] Sentence-BERT: Sentence Embeddings using Siamese BERT-Networks, Nils Reimers and Iryna Gurevych

---

> > > ### Author Response · Authors · 2019-11-15
> > > **Response to Review #3**
> > >
> > > Thank you for the additional questions.
> > >
> > > 1: MLModelScope is a platform design to evaluate and benchmark trained ML models. The MLModelScope design, as presented, does not address training. Scalable model evaluation or scalability in the paper does not mean training a single ML model with multiple systems but means evaluating ML inference across different models/frameworks/systems in parallel using multiple systems.
> > >
> > > 2,3: The "fair and reproducible" benchmarking is guaranteed by the design (the main contribution of the paper). When training a model, we agree that the model author specifies the order of preprocessing stages in code, but this is not where MLModelScope is used. When one wants to compare different trained models and systems (e.g. to select the best combination to deploy an ML task) they have to have a  “fair and reproducible” way of performing the evaluation and comparison of all the available options. Allowing for a scalable way to perform this evaluation allows one to cope with the mirage of models and systems that are proposed daily.
> > >
> > > Therefore, it is inference/evaluation, and not training, where MLModelScope aims to help. This is recognized by the other two reviewers who agree that MLModelScope provides a unified approach to specify, evaluate, and benchmark ML across models, software stacks, and systems. The evaluation section serves as a case study and showcases example experiments or comparisons that can be performed through MLModelScope in terms of models, hardware, and frameworks. We sincerely ask the reviewer to consider the use case of MLModelScope — evaluating and benchmarking trained models —-  and compare us against related work.
> > >
> > > 4. Again, MLModelScope is for evaluating and benchmarking trained ML models to help ML deployment, not training. Since the models evaluated do not use non-deterministic layers, we did not consider random number generation in our evaluation. MLModelScope's design does not preclude one from creating model manifests for models (trained using different scenarios such as random seeds), fixing the random seed within the pre-processing code section in the manifest, and leveraging MLModelScope to compare the performance or accuracy across these models. Section 3 describes this in detail.
> > >
> > > We truly think the reviewer has misunderstood the design and the scope of the paper. Please let us know if there’s anything else that we can make clear within the paper that would clarify the confusion.

---

> > > > ### Comment · AnonReviewer4 · 2019-11-15
> > > > **scalability novelty**
> > > >
> > > > I would like the author to clarify the difference of their model evaluation versus  using "distributed" methods already available in deep learning framework, Lets assume there is a model written with pytorch, my argument is that this is already provided to quickly make this model evaluation in a distributed way. Not only pytorch by most of the deep learning framework allow this distributed evaluation.
> > > >
> > > > In your framework, you call the model part of the methods, "inference" part, and evaluate it in a distributed way, I appreciate the author clarify how this is difference from using the distributed way of doing this evaluation already provided in deep learning frameworks?

---

> > > > > ### Author Response · Authors · 2019-11-15
> > > > > **Response to Review #3**
> > > > >
> > > > > Thank you for the additional questions.
> > > > >
> > > > > The "distributed" methods in deep learning frameworks are for training a single ML model with multiple systems. Using these "distributed" methods is not valid for deployment and models trained with multiple systems are deployed on a single system. For example, you can train a BERT model using thousands of GPU systems, but in the end, you deploy it on a single system (either GPU or CPU or TPU), otherwise, it’s a huge resource waste. In summary, inferring model in deployment is a different process from training a model. There are standalone libraries such as TensorFlow Serving or MXNet Serving (which are not within TensorFlow or MXNet) to serve model inference.
> > > > >
> > > > > MLModelScope is a platform to evaluate and benchmark trained ML models for ML deployment. MLModelScope has a distributed runtime, meaning it can evaluate ML models across different models/frameworks/systems in parallel using multiple systems.  For example, you can evaluate 100 different BERT models at the same time using 100 systems. Note that this is different from training where a single model is trained in parallel across many systems. A system design that has a distributed runtime to perform an evaluation at scale and provides a specification to ensure repeatable and fair evaluation did not exist until MLModelScope.

---

> > > > > > ### Comment · AnonReviewer4 · 2019-11-15
> > > > > > **comments on model**
> > > > > >
> > > > > > * I do think one can easily deploy the model and evaluate them in a distributed way using Pytorch/tensorflow.
> > > > > > * could you clarify what do you mean by evaluating on different frameworks and systems?
> > > > > >
> > > > > > * One more import issue is that I would like to clarify is that I think you assume the results of deep learning models could change if they are evaluated on the different SW/HW stack, right? My question is that how did you come to this conclusion? By evaluating the same code on different HW/SW normally one would not observe any differences, let's assume seed is fixed, why do you assume in the first place SW/HW matter in the evaluation of deep learning models?

---

> > > > > > > ### Author Response · Authors · 2019-11-15
> > > > > > > **Response to Review #3**
> > > > > > >
> > > > > > >
> > > > > > > * That is not true. PyTorch or TensorFlow are just frameworks to execute a model graph for inference or training. They do not serve or manage models and no runtime to handle multiple models at the same time exist in them. MLModelScope wraps frameworks such as  PyTorch or TensorFlow into framework predictors (See Section 3.2 in the paper), and which are coordinated and managed by the MLModelScope runtime to perform model evaluation tasks. Please take a look at  https://pytorch.org/blog/model-serving-in-pyorch/. TensorFlow Serving
> > > > > > > https://github.com/tensorflow/serving and MXNet Model Server https://github.com/awslabs/mxnet-model-server mentioned in the authors’ previous comment are the model servers mentioned in the blog post.
> > > > > > >
> > > > > > > MLModelScope is for evaluation and benchmarking is framework agnostic, it shares some common design features as the model servers and model microservices, this is detailed in Section 3.2 and Appendix A.3.
> > > > > > >
> > > > > > > * The authors do believe the reviewer only considered the accuracy or quality of models when raising the questions or making the statement “By evaluating the same code on different HW/SW normally one would not observe any differences”. Accuracy or quality of models is just one aspect of model evaluation, performance such as latency, throughput, memory usage, etc. is another important aspect and is often the key factor in model selection when building ML applications.
> > > > > > >
> > > > > > > There is a performance difference when evaluating models using different frameworks/hardware, as shown in Figure 8. And there has been a significant drive by the ML community and industry to publish reference model benchmarks for evaluation (e.g. MLPerf https://mlperf.org/inference-results).

---

> > > > > > > > ### Comment · AnonReviewer4 · 2019-11-15
> > > > > > > > **response to authors**
> > > > > > > >
> > > > > > > > Dear Authors,
> > > > > > > >
> > > > > > > > I checked the resources, when I was reading the paper, I did not see multiple model evaluation being highlighted, but now I understand your point of handling multiple models, thanks for sharing the links.
> > > > > > > >
> > > > > > > > I also agree with your point on performance differences, though I still think in terms of accuracy HW/SW does not contribute much, I think in the paper, you highlight "fair evaluation" and also "accuracy", which could bring possible confusions but I agree with your point in terms of performance.
> > > > > > > >
> > > > > > > > I still think first three pages of experiments on preprocessing are not relevant to this work.
> > > > > > > > Experiment in 4.2 is also written in a way that you want to find if CPU/GPU instances are more efficient, which I also did not find to be relevant either.
> > > > > > > >
> > > > > > > > However, I highlight I did not assess the model design, which is the main contribution of the paper, and  did not know the background of prior work to really assess the novelty of the work, my score is solely based on the experiments. Additionally I changed my response to the assessment of the derivation and theory to N/A to highlight it.

---

> > > > > > > > > ### Author Response · Authors · 2019-11-15
> > > > > > > > > **Response to Review #3**
> > > > > > > > >
> > > > > > > > > Thank you for the prompt response.
> > > > > > > > >
> > > > > > > > > Since MLModelScope is a tool, we wrote the evaluation section as a case study to show example experiments or comparisons that can be performed using MLModelScope. The results are not the focus but the capability of getting those results through MLModelScope is what we have tried to show. We also linked to a video demonstrating the web UI of the platform in the Appendix of the paper.
> > > > > > > > >
> > > > > > > > > The paper is submitted to the Tools/Systems area of the conference, which judges the paper based on its design and utility to the ML community. We do not think the reviewer is arguing that a platform, such as MLModelScope, would not be useful so long as the design is sound (as the other reviewers confirmed). And, unlike algorithm papers where models or methods are proposed, a platform design does not have a simple metric such as “improves the accuracy by X%” or “achieves a speedup Nx”. Instead, a platform should be evaluated based on what it enables. Thus, we think presenting the capabilities of the platform should be the focus of the evaluation section, and we have tried to do so. We will clarify the purpose of the evaluation section of the paper. If the reviewer has suggestions on what else should be shown in the evaluation section to demonstrate such as platform design, then we are open to such suggestions.
> > > > > > > > >
> > > > > > > > > Since the paper is submitted to the Tools/Systems area of the conference, which judges the paper based on its design and utility to the ML community, we sincerely hope the reviewer can reconsider the “Rating” after re-evaluating the design of the platform and evaluate the experiment section based on its enablement capabilities. Thank you.

---

> > > > > > > > > > ### Comment · AnonReviewer4 · 2019-11-15
> > > > > > > > > > **experiments**
> > > > > > > > > >
> > > > > > > > > > The main issue with this paper from my view is that 3 pages of the experiments on our preprocessing, which is not relevant to the point of this paper, the authors are proposing a general framework for evaluation of machine learning models, and they argue that this is scalable ... , however, I cannot see much experiments on showing its scalability, what is the bottleneck of the system, how many models it can handle, how large each model can be? On 4.2, you only consider an "Inception-v3 model", with 9 instances, but the argument of the paper is quite general, and the authors argue for scalability, I, however, don't see experiments backing up this argument much.

---

> > > > > > > > > > > ### Author Response · Authors · 2019-11-15
> > > > > > > > > > > **Response to Review #3**
> > > > > > > > > > >
> > > > > > > > > > > Being able to conduct model evaluation at scale is one of the features of the design and follows from how the system is architected.
> > > > > > > > > > >
> > > > > > > > > > > As stated in our previous comments, MLModel leverages a microservice architecture design (e.g. as described in https://pytorch.org/blog/model-serving-in-pyorch/) where each evaluation is done in a process (framework predictor) on a system. MLModelScope is inherently scalable because any number of framework predictors (i.e. services) and models (via model manifest) can be registered in the registry (e.g.https://www.consul.io/) to perform evaluation and there is no interaction between the framework predictors (no interaction between any two services).   MLModelScope enables model evaluation at scale and we think it is self-explanatory given the runtime design and its flow (as presented in Sections 3.2 and 3.3). The registry is dynamic; i.e. predictors can be registered at runtime thus enabling an elastic design based on the workload. MLModelScope uses an orchestration layer (Figure 2) to load balance requests across the registered predictors.
> > > > > > > > > > >
> > > > > > > > > > > Thus, we focus on showing the example experiments through MLModelScope in the evaluation section. We encourage the reviewer to read through this high level description of microservice and service discovery (https://en.wikipedia.org/wiki/Microservices and  https://docs.aws.amazon.com/whitepapers/latest/microservices-on-aws/service-discovery.html ) to better understand the distributed runtime and look into Section 3.2 which describes the runtime design of MLModelScope.

---

### Decision · Program_Chairs · 2019-12-19

**Decision:**

Reject

**Comment:**

The paper proposes a platform for benchmarking, and in particular hardware-agnostic evaluation of machine learning models. This is an important problem as our field strives for more reproducibility.

This was a very confusing paper to discuss and review, since most of the reviewers (and myself) do not know much about the area. Two of the reviewers found the paper contributions sufficient to be (weakly) accepted. The third reviewer had many issues with the work and engaged in a lengthy debate with the authors, but there was strong disagreement regarding their understanding of the scope of the paper as a Tools/Systems submission.

Given the lack of consensus, I must recommend rejection at this time, but highly encourage the authors to take the feedback into account and resubmit to a future venue.